# Utilizing Constant Energy Difference between sp-Peak and C 1s Core Level in Photoelectron Spectra for Unambiguous Identification and Quantification of Diamond Phase in Nanodiamonds

**DOI:** 10.3390/nano14070590

**Published:** 2024-03-27

**Authors:** Oleksandr Romanyuk, Štěpán Stehlík, Josef Zemek, Kateřina Aubrechtová Dragounová, Alexander Kromka

**Affiliations:** 1Institute of Physics of the Czech Academy of Sciences, Cukrovarnická 10, 162 00 Prague, Czech Republic; stehlik@fzu.cz (Š.S.); zemek@fzu.cz (J.Z.); dragounova@fzu.cz (K.A.D.); kromka@fzu.cz (A.K.); 2New Technologies—Research Centre, University of West Bohemia, Univerzitní 8, 306 14 Pilsen, Czech Republic; 3Faculty of Nuclear Sciences and Physical Engineering, Czech Technical University in Prague, Břehová 7, 115 19 Prague, Czech Republic

**Keywords:** nanodiamonds, hydrogenation, XPS, UPS, Raman spectroscopy, surface graphitization, Ar cluster ion beam sputtering

## Abstract

The modification of nanodiamond (ND) surfaces has significant applications in sensing devices, drug delivery, bioimaging, and tissue engineering. Precise control of the diamond phase composition and bond configurations during ND processing and surface finalization is crucial. In this study, we conducted a comparative analysis of the graphitization process in various types of hydrogenated NDs, considering differences in ND size and quality. We prepared three types of hydrogenated NDs: high-pressure high-temperature NDs (HPHT ND-H; 0–30 nm), conventional detonation nanodiamonds (DND-H; ~5 nm), and size- and nitrogen-reduced hydrogenated nanodiamonds (snr-DND-H; 2–3 nm). The samples underwent annealing in an ultra-high vacuum and sputtering by Ar cluster ion beam (ArCIB). Samples were investigated by in situ X-ray photoelectron spectroscopy (XPS), in situ ultraviolet photoelectron spectroscopy (UPS), and Raman spectroscopy (RS). Our investigation revealed that the graphitization temperature of NDs ranges from 600 °C to 700 °C and depends on the size and crystallinity of the NDs. Smaller DND particles with a high density of defects exhibit a lower graphitization temperature. We revealed a constant energy difference of 271.3 eV between the sp-peak in the valence band spectra (at around 13.7 eV) and the sp^3^ component in the C 1s core level spectra (at 285.0 eV). The identification of this energy difference helps in calibrating charge shifts and serves the unambiguous identification of the sp^3^ bond contribution in the C 1s spectra obtained from ND samples. Results were validated through reference measurements on hydrogenated single crystal C(111)-H and highly-ordered pyrolytic graphite (HOPG).

## 1. Introduction

Nanodiamonds (NDs) have potential applications in a wide range of technologies that exploit their unique electronic and surface properties. Their crystalline diamond core imparts structural stability and biocompatibility, while the ability to tailor surface chemistry and functional groups allows controllable interaction with various systems. As a catalytically active material, NDs show promise for photo- and electrocatalytic reactions like nitrogen reduction [1] or hydrogen production [2]. They may also find use in energy storage as electrode materials in batteries and supercapacitors [3]. Their tunable electronic bandgaps and band edge positions relative to vacuum through modification of size, defects, non-diamond carbon phases, and surface chemistry [4] could enable applications in optoelectronics such as light-emitting diodes, photovoltaics, and electron emitters [5]. Biomedical applications [6] are envisioned as well, taking advantage of biocompatibility and the broad options to functionalize nanodiamond surfaces for drug delivery or imaging.

Several types of nanodiamonds are commonly studied and utilized, mainly differentiated primarily by their production method and the resulting structural characteristics. Detonation nanodiamonds (DND) are synthesized via controlled explosions of carbon-containing explosives and are the most abundantly available type. Due to their explosive formation, detonation nanodiamonds tend to be sub-10 nm in size and highly polycrystalline and roundish in shape. They also often contain a significant portion of non-diamond carbon impurities incorporated during their preparation process [7]. In contrast, high-pressure high-temperature (HPHT) nanodiamonds [8] are produced by milling of synthetic diamond microcrystals grown by the HPHT process. HPHT NDs are generally larger than DND NDs, with particles exhibiting uniform monocrystalline and highly irregular shapes. HPHT NDs also demonstrate reduced and surface-limited non-diamond sp^2^ carbon phase content relative to DNDs [9].

Realization of the full potential of NDs requires functionalization and understanding of the properties of ND surfaces. Basic surface chemistry modifications (oxidation, hydrogenation) can be achieved by convenient thermal treatments which can be followed by more complex chemical functionalization required for specific uses [4,10]. Like for diamond single crystals, surface hydrogenation of HPHT NDs causes the true negative electron affinity (NEA) [4] making NDs an efficient photoemitter [11], and facilitates disintegration of DNDs [9,12].

X-ray Photoemission Spectroscopy (XPS) is a suitable method for the investigation of chemical and electronic properties of ND surfaces, particularly to reveal the surface composition and chemical bond types, to assess the sp^2^/sp^3^ carbon ratio, and to specify the surface terminations or changes in the surface electronic properties of NDs [4,13,14,15,16]. However, there are several partially solved or unsolved problems in the analysis of ND surfaces. Diamond is a wide-bandgap material (5.5 eV) and therefore highly electrically resistive. Its surface becomes positively charged during measurements, causing recorded spectra to shift towards higher binding energy (BE). In the case of homogeneous and time-stable surface charging, the spectra are shifted by a constant value. To compensate for the surface charging, current spectrometers are usually equipped with low-energy flood guns. Precise compensation and spectra calibration, however, is sometimes not straightforward.

One can quickly detect surface charging by changing the input power of the X-ray source [16] or, more precisely, by applying a negative bias voltage to the sample holder [14]. The use of flood guns can easily lead to under- or over-compensation of the analyzed surface [17]. However, in diamond samples, selecting an appropriate internal standard poses a challenge. For this purpose, the C 1s peak maximum is usually employed [18], along with the C sp^2^ contribution located at the position of the C 1s line recorded from graphite [19], or the O 1s peak position [20]. Regardless of the uncertainty associated with this correction, the described referencing is affected simultaneously by the presence of surface upward or downward band bending due to the surface graphitization and hydrogenation of NDs [14]. Moreover, the analysis of heterogeneous samples often leads to heterogeneous surface charging [21,22] and such analysis can be tricky or even impossible.

The sp^3^-hybridized bonds contribution in C 1s peaks are identified as the highest intensity component at around 285.0 eV (after spectra calibration) for diamonds [12,23]. Density functional theory, as utilized by Fujimoto et al. [21] supports the assignment of the sp^3^ contribution at a lower binding energy of the C 1s line. The authors argue that the conflicting assignment is a result of surface charging of diamond. Haerle et al. [24] and subsequently Titantah et al. [25] employed first-principles calculations to determine a core-level position and shift between the sp^3^ and sp^2^ contributions for amorphous carbon or carbon-based materials, yielding a value of ~1 eV, independent of mass density of carbon phase. Utilizing this value, the C sp^2/^C sp^3^ phase ratios align with measurements of the sp^2^/sp^3^ component areas in the C 1s line [24] and with the findings obtained by high-energy electron energy-loss spectroscopy (HREELS) [25]. This assignment gains further support from experimental investigations on crystalline diamonds annealed at high temperatures under ultra-high vacuum conditions [12,26], as well as from studies about interaction of energetic ions with diamond surfaces [19,27]. A progressively increasing sp^2^ bond-related component at the low binding energy side of the C 1s peaks as temperature and/or ion dose increase can serve as indicator of diamond graphitization process.

The line shape of C 1s spectra recorded from oxygen-free diamond surfaces sometimes exhibits asymmetry. In other words, there is an additional component shifted positively by 0.5–0.9 eV with respect to the sp^3^ component in C 1s spectrum. Hence, this spectral intensity cannot be attributed to carbon bonded to oxygen. In the existing literature, it is attributed to photoemission from carbon atoms situated within various defects [28,29,30], from hydrogen bonded to carbon atoms, denoted as C-H_x_ [15], where x ≥ 2 [19,31,32]. High intensity of this component was observed in the C 1s spectra of DND samples [29,33]. Considering this complexity, calibrating the peak maxima of the C 1s line by assigning either the positions to the sp^2^, sp^3^ components or to the sp^3^, C-H_x_ components is not straightforward due to the similarity of the chemical shifts in both cases. For the correct assignment of the individual C 1s peak components, the initiation of the phase transition between the C sp^3^ phase of diamond and the C sp^2^ graphite phase, could help to resolve the problem.

The graphitization of monocrystalline diamond surfaces [34], nanocrystalline diamond (NCD) films [35], and DNDs [29] during annealing under ultra-high vacuum (UHV) conditions has been previously studied. The temperature required to start the surface graphitization reaction on a monocrystalline diamond (111) surface under UHV conditions was reported to be approximately 850–900 °C [34]. A similar temperature range for graphitization was observed for DND particles: below 900 °C, the ND surface undergoes reconstruction into isolated graphitic domains, evident from the emergence of the sp^2^ carbon peak as observed by XPS, while the diamond core remains unaltered. Bulk graphitization of ND cores occurs above 900 °C. It should be noted that significantly lower graphitization temperatures (<500 °C) were observed in the presence of a Ni atoms (catalyst layer) on a NCD surface [35], i.e., graphitization temperature also depends on the diamond surface chemistry.

The graphitization of NCD films can also be reached through monoatomic Ar ion sputtering and Ar cluster ion beam (ArCIB) sputtering [19,36], leading to an increase in concentration of the C sp^2^ phase on the NCD surface. In contrast to monoatomic Ar ion sputtering, no Ar atoms were incorporated into the NCD film using the ArCIB sputtering. Thus, no interaction between Ar ions and C atoms [37], nor formation of Ar-C bonds [36] are expected by using ArCIB sputtering.

The recent investigation of graphitization on DND-H samples through laser irradiation was reported [33]. The study indicated an increase in the concentration of the C sp^2^ phase with longer irradiation times, while the concentration of defects or C-N bonds remained nearly constant. Although an exact temperature was not provided, the results suggest that laser irradiation induced a localized temperature increase of the DND samples, which was adequate to initiate graphitization of the surface layers of DND. This temperature increase is likely below 900 °C, as suggested by previous thermal studies [29].

Photoemission from the valence bands (VB) of diamonds [38,39,40] and NDs [14,15,41] reveals spectral features originating from the joint sp-electron and p-electron partial density of states [38]. The sharp sp-peak between 12–14 eV has been primarily attributed to carbon sp^3^ bonds [39,42]. Graphitization of the microcrystalline diamond film, however, results in a reduction of the sp-peak intensity, but the peak intensity is not completely eliminated for the graphitized carbon [41] or graphene [43]. Therefore, the use of C sp^3^ notation for the sp-peak could be misleading. On the other side, the position of the sp-peak could be used for the calibration of the VB spectra measured from diamond films with a mixture of C sp^2^/C sp^3^ phases [44].

In the present study, we examined the graphitization of three different types of hydrogenated NDs using XPS, UPS, and Raman spectroscopy. For the calibration purposes, we employed reference samples of hydrogenated single crystal diamond C(111)-H and freshly cleaved HOPG. The ND samples were subjected to annealing in UHV and sputtering using an ArCIB. We also establish a correlation between the sp-peak position in the VB and the position of the sp^3^ component in the C 1s line and employ it for the calibration of photoelectron spectra from ND samples.

## 2. Experimental Details

As HPHT NDs, we used monocrystalline synthetic NDs from Pureon (Lengwil, Switzerland), featuring a size range of 0–30 nm. To reduce the non-diamond carbon content, the as received HPHT NDs were subjected to annealing in ambient air atmosphere at 450 °C for 5 h, resulting in purified and oxidized HPHT NDs [9]. Next, the HPHT NDs were annealed in hydrogen gas at a flow rate of 5 L/min at atmospheric pressure for 6 h, at an annealing temperature of 800 °C [45]. This sample is further labeled as *HPHT ND-H*.

As DNDs we used two different materials. First we used conventional DNDs from New Metals and Chemicals Corp. (Tokyo, Japan) with a nominal size of about 5 nm and 2 at.% of nitrogen [46]. The as-received DNDs were first annealed in ambient air atmosphere at 450 °C for 30 min and then hydrogenated in hydrogen gas at a flow rate of 5 L/min at atmospheric pressure for 6 h at an annealing temperature of 700 °C. This sample is further labeled as *DND-H*.

The second DND material was provided by OZM Research Ltd. (Hrochův Týnec, Czech Republic), featuring a nominal size of approximately 2–3 nm [estimated by transmission electron microscopy (TEM) and small-angle X-ray scattering (SAXS) measurements, not shown here] and a reduced nitrogen content as shown below. The particles were first annealed in air at 350 °C for 30 min and then hydrogenated at 700 °C for 3 h at a flow rate of 5 L/min at atmospheric pressure of hydrogen. This sample is further labeled as *snr-DND-H* where *s*, *n*, *r* stand for size-nitrogen-reduced NDs.

ND powders were first dispersed in water (1.0 mg of ND powder in 1.0 mL of deionized water) by sonication using a rod-type sonicator (Hielscher UP200s, Teltow, Germany) at 200 W for 1 h. The ND colloids were drop casted on gold bulk substrates and dried before loading in UHV chamber for XPS and UPS measurements. We ensured sufficient thickness of ND layers on the conducting substrate. The as-prepared ND samples (for measurements at room temperature, RT samples) were transparent and became non-transparent after annealing, qualitatively signalizing increase of their C sp^2^ content [see Appendix A].

Monocrystalline diamond C(111) and HOPG samples were used as reference of C sp^3^ and C sp^2^ phase materials. In order to clean C(111) surface and to obtain a hydrogenated diamond surface, a monocrystalline C(111) sample was exposed to a 10 min hydrogen plasma treatment in a multimode clamshell cavity reactor (SDS6K, Seki Diamond Systems, Seki City, Japan). The process conditions were as follows: H_2_ gas flow rate of 300 sccm, pressure of 4 kPa (30 Torr), microwave power of 2 kW, and the average process temperature of 465 ± 5 °C. The contamination-free surface of the HOPG was prepared through exfoliation. The process involved attaching the 10 × 10 mm^2^ bulk HOPG sample to the sample holder. Scotch tape was then applied multiple times to remove the topmost contaminated layers. Once the largest area of the HOPG remained on the scotch tape, the bulk sample with freshly-prepared surface was promptly inserted into the spectrometer chamber.

Raman spectra were measured using a Renishaw InVia Reflex Raman spectrometer (Renishaw, Wotton-under-Edge, UK) with an excitation wavelength of 442 nm (Dual Wavelength He–Cd laser, model IK5651R-G, Kimmon Koha, Tokyo, Japan) and air-cooled CCD camera. The samples were exposed by a continuous wave laser (the intensity power of 0.33 mW) focused to a spot of 1 μm exposure time was set for 100 s, and spectra were sampled 42 times. Spectra were measured in a backscattering setup with a 100× objective (Leica, Wetzlar, Germany, NA = 0.9) in the confocal mode with a grating of 2400 lines/mm.

XPS spectra were acquired with an AXIS-Supra photoelectron spectrometer (Kratos Analytical Ltd., Manchester, UK) using monochromatized Al Kα radiation (1486.6 eV, 150 W, analyzed area of 0.7 × 0.3 mm^2^). The X-ray incidence angle and photoelectron emission angle were 36° and 90°, respectively. The high-resolution core level spectra were recorded with a pass energy of 10 eV and with an overall energy resolution of 0.45 eV. The valence band XPS spectra (VB-XPS) should be measured with the same pass energy to avoid possible charge-related shifts for non-conductive samples. Low energy electron flood gun was used for neutralization of charge on surfaces. The higher-resolution spectra were measured with an energy step of 0.1 eV.

Sample annealing up to 800 °C was carried out in situ in an XPS chamber with temperature increments of 100 °C. After each annealing step, in situ XPS spectra were acquired at a given temperature. Throughout the annealing process, the pressure in the XPS chamber remained below 2.0 × 10^−7^ Torr. To maintain UHV conditions, the sample temperature increased gradually. It took approximately 20–40 min to raise the temperature by 100 °C, and a similar period was needed to acquire the photoelectron spectra.

Ar cluster ion beam (ArCIB) sputtering was conducted on annealed samples at room temperature, following the cooling of the samples from 800 °C to room temperature (more than 4 h was needed for cooling to room temperature). Sputtering was performed with the Kratos Gas Cluster Ion Source (Minibeam 6) with a cluster size of 1000 atoms and the Ar ion beam energy of 5 keV. The exposure area of the ArCIB was set to 2 × 2 mm^2^, and the incident angle of the ion beam was set to 45° from the surface normal. The sample surfaces underwent sputtering for a duration of 60 min. The UPS spectra were acquired utilizing He II light excitation (40.8 eV) with a pass energy of 40 eV and an energy step size of 0.1 eV. Note, the He I was less suitable for resolving sp-peak in the valence spectra because it overlapped with the secondary electron emission intensity. Therefore, He II with a broader valence spectra range was used.

Quantification of atomic composition was carried out by analyzing the integrated high-resolution peak areas, following Shirley’s electron inelastic background subtraction. We employed the ESCApe software (Kratos Analytical Ltd., ver. 1.4.0.1149), utilizing the incorporated atomic sensitivity factors, and employed the KolXPD software (ver. 1.7.0.8) for the core-level peak treatment [47]. Fitting of the XPS and UPS spectra was carried out by Voigt and Gaussian functions, respectively. The width of the Lorenzian function in the Voigt function was consistently maintained in all spectra.

## 3. Results and Discussion

Prior to analyzing the ND samples, photoelectron spectroscopy measurements were carried out on the reference C(111)-H and HOPG samples, with defined majority of the C sp^3^ and C sp^2^ phases, respectively. Figure 1 displays the measured high-resolution C 1s spectra, valence band XPS spectra (VB-XPS), and UPS He II spectra from the C(111)-H and HOPG samples. The reference sample surfaces were relatively clean: an O concentration of 0.8 at.% was measured on the C(111)-H surface and no O was present on HOPG surface. The position of the C 1s peak of C(111)-H sample was calibrated to 285.0 eV. The VB-XPS spectrum in Figure 1c underwent the same calibration binding energy shift as for C 1s line. For HOPG spectra in Figure 1b,d, no calibration was needed (i.e., no surface band bending was observed, the as-measured XPS and UPS spectra are presented in figure). The binding energy of the sp^2^ component in HOPG was at 284.4 eV. The C 1s spectrum of the C(111)-H sample was deconvoluted into components with the chemical shifts of −0.8 eV (for sp^2^ component), 0.5 eV (for C_x_ component), and 1.1–1.5 eV (for C-O/C-OH) relative to the position of the sp^3^ component. The C 1s spectrum of HOPG sample was fitted by one sp^2^ (asymmetrical, Doniach-Sunjic) component, confirming clean and well-ordered HOPG surface.

In previous studies, eight components were utilized for fitting VB-XPS spectra of diamond and HOPG samples [44,48]. In this study, we used a similar component distribution in the VB spectra but with additional components near the sp-peak to capture the shape of the peak more accurately. The sp-peak maximum position was at 13.7 ± 0.1 eV for both samples. We should emphasize, however, that the measured position of peaks could be affected by surface band bending [11], but the relative energy difference between C 1s and sp-peak should not. The measured energy differences between the C 1s peak components and the sp-peak positions were 271.3 ± 0.1 eV (for sp^3^) and 270.7 ± 0.1 eV (for sp^2^). These relative binding energy differences were used for identification of the sp^3^ and sp^2^ components in the C 1s spectra from ND samples.

The UPS He II spectra (after the background subtraction) are depicted in Figure 1e for the C(111)-H sample and Figure 1f for the HOPG sample. The sp-peaks were present on both diamond and HOPG UPS spectra [44] despite of the absence of C sp^3^ phase in HOPG [see Figure 1b]. The UPS spectra were then calibrated [44] to the position of the sp-peak at 13.7 eV (note, charge-induced shifts could be different for XPS and UPS spectra because of difference in excitation energies). Unlike the VB-XPS spectra, the UPS spectra exhibit an amplified contribution from the p-electrons within the range of 0–11 eV [38,39]. The π-band peak at 3.6 eV is clearly discernible in Figure 1f for the HOPG sample. So, an intensified π-band peak indicates the presence of an C sp^2^ phase on the diamond on a surface.

In Figure 2a–c the C 1s peaks from HPHT ND-H, DND-H, and snr-DND-H samples are displayed. The corresponding survey spectra and N 1s spectra from DND samples are included in Appendix A. The spectra were measured from as-prepared samples at room temperature (RT, red), from annealed samples at 800 °C (green), and from sputtered samples (ArCIB at RT, blue). All spectra were calibrated with respect to the position of the sp^3^ component at 285.0 eV. Fitted components are shown for RT samples exclusively. The core level shifts and component assignments to bond types are similar to those of the reference sample in Figure 1 and in agreement with our prior studies [4,9,19,33]. The C_x_ component of DND-H sample spectra also involves contributions from C-N bonds. The concentration of C-N bonds is small, however, and therefore these components were not resolved by the fit. The values of the chemical core level shifts and the measured atomic compositions of NDs are included in Appendix A.

A negative chemical shift of −1.5 eV was obtained for the sp^2^ component of sputtered DND-H samples (with 2.0–2.5 at% of N atoms in DND-H). Smaller shifts of −0.8 eV and −0.9 eV were derived for snr-DND-H samples. Similar shifts of about −1.4 eV for the sp^2^ components were previously measured on graphitized DND samples with N contaminants [29] and on defective N-doped graphene layers [49]. Shifts were identified as due to hybridization of C sp^2^ bonds close to vacancies [49]. This could correlate with sputtering of DND-H samples (introducing vacancies in C sp^2^ phase) in a proximity of N atoms.

Examination of the C 1s peaks disclosed partial graphitization of the sample surfaces: following sample annealing, the areas of sp^2^ components exhibited slight augmentation, and a more notable increase in the sp^2^ component area was observed after ArCIB sputtering, where more pronounced amorphization of NDs and increase of sp^2^ bond concentration is expected. After sputtering, sample surface morphology has also changed, specifically for HPHT ND-H (see Appendix A). The increase of the sp^2^ component area varied among different types of ND particles, affirming the link between graphitization efficiency and NP types (or size): the concentration of the graphitic phase was higher after annealing and sputtering for the smaller and more defective DND samples.

Figure 2d–f show VB-XPS spectra of RT (as-received), annealed, and sputtered samples. The VB spectra of the as-received HPHT ND-H sample [in (d), red dots] are very similar to the VB spectra of the diamond monocrystal sample [Figure 1c] because the emission from the diamond core dominates and the C sp^2^ phase concentration is small. The sp-peaks are broader in RT DND-H and snr-DND-H VB spectra [Figure 2e,f, red dots], where concentration of C sp^2^ phase is higher. A similar broadening was observed between the C(111)-H and HOPG samples in Figure 1c,d. The shape of the sp-peak therefore depends on the concentration of C sp^2^ and C sp^3^ phases. It is important to note that similarly to the reference bulk samples, the binding energy difference between the sp^3^ components and the sp-peaks was preserved (271.3 ± 0.1 eV for all as-prepared ND samples).

Annealing and sputtering induced further broadening of sp-peaks [green and blue curves in Figure 2d–f], but the sp-peak was still preserved and no contribution from the π-band peak was resolved on the graphitized HPHT ND-H samples [Figure 2d]. More prominent graphitization has occurred for smaller DND-H and snr-DND-H particles. In particular, the sp-peaks have smeared out and the π-peak at around 3.6 eV appeared on the VB-XPS spectra [Figure 2e,f].

Figure 3 illustrates the dependence of the sp^2^/sp^3^ component area ratios on annealing temperature as derived from the C 1s spectra analysis. Initial concentration of C sp^2^ phase on as-prepared HPHT ND-H sample was smaller than for DND samples. Annealing up to 700 °C did not alter the C sp^2^/C sp^3^ ratio significantly. At >700 °C, graphitization starts, leading to an increase in sp^2^/sp^3^ phase concentration ratio.

The initial concentration of the C sp^2^ phase was higher for DND and snr-DND-H samples, which is consistent with the higher concentration of contaminations observed for DND samples (see Appendix A). Up to 600 °C, the sp^2^/sp^3^ ratio remained below ~0.1 for DND-H samples but decreased for snr-DND-H samples. In the latter cases, oxygen desorption could be associated with a reduction in the C sp^2^ phase on the surface. After 600 °C, the sp^2^/sp^3^ ratio increased for both types of DND samples. Note, the sp^2^/sp^3^ concentration was higher for the smallest snr-DND-H particles than for DND-H particles.

In terms of DND, our results are comparable to previous study, where graphitization temperatures of >700 °C for DND surfaces and >900 °C for DND cores have been reported [29]. Here, we show that the surface rearrangement of DND-H starts already at the temperature > 600 °C. We assign this minor difference to a slight difference in ND particle size and crystallinity of NDs. In contrast to DND, the HPHT ND-H surface rearrangement starts at the higher temperature > 700 °C. The surface rearrangement eventually leading to the ND graphitization is triggered by the thermal desorption of the surface functional groups, which is accompanied by the formation of surface dangling bonds. In vacuum this leads to the formation of the sp^2^ bonds between the adjacent carbon atoms and formation of C-sp^2^ patches. However, in atmosphere, the surface dangling bonds can be passivated by surrounding atoms such as hydrogen [50]. Indeed, the observed graphitization onset temperature match well with the temperature of thermal hydrogenation, i.e., 600–700 °C for DND [51] and 700–800 °C for HPHT ND [15,52]. Our findings confirm that the initial graphitization temperature of ND depends on the type and size of studied NDs, i.e., the temperature is lower for smaller and more defective DNDs.

Figure 4 displays UPS He II spectra measured on (a) HPHT ND-H, (b) DND-H, and (c) snr-DND-H samples at room temperature (RT, red line) and after annealing in UHV (800 °C, green line). The UPS spectra were calibrated to the sp-peak positions (and π-band positions). Due to the higher surface sensitivity of UPS compared to VB-XPS, the enhancement of the π-band was more clearly distinguishable, especially for HPHT ND-H samples. The sp-peak deteriorated with the decrease in NP size (resulting in a lower concentration of C sp^3^ core phase) and with graphitization (leading to an increase in C sp^2^ phase on NP surfaces).

In Figure 4c, the π-band peak was well-resolved and sp-peak was very weak on as-prepared snr-DND-H sample at RT (with the highest C sp^2^ phase concentration). After annealing, the intensity of the π-band peak increased, and the sp-peak completely disappeared. We can assume that the surface of the snr-DND-H sample was almost completely covered by the C sp^2^ phase layer.

The appearance of the π-band peak and variation in the sp-peak intensities with sp^2^/sp^3^ ratio change in NDs agree with the previous results obtained for the diamond films [41,44,48]. Therefore, we would like to emphasize that a combination of XPS, VB-XPS, and UPS measurements has a great potential for the spectra calibration and resolving different phases of carbon films and diamond NPs.

Figure 5 shows Raman spectra of as-prepared and sputtered (a) HPHT ND-H, (b) DND-H, and (c) snr-DND-H samples measured by 442 nm excitation laser. The diamond peak in HPHT ND-H spectra at 1332 cm^−1^ is very sharp (blue rectangular), confirming the dominated C sp^3^ core structure of as-prepared HPHT ND-H sample (measured at RT, red line). In as-prepared DND-H, the diamond peak is broader and shifted to lower wavenumbers (1327–1328 cm^−1^) compared to bulk diamonds. A weaker and broader diamond peak at about 1340 cm^−1^ was also resolved on snr-DND-H RT sample, but the spectrum is rather noisy due to very small ND particle size (the diamond sp^3^ peak of the snr-DND-H sample could be better resolved on UV-Raman spectrum, see Appendix A). In accordance with the XPS results the C sp^2^ carbon phase content increases in the line HPHT ND-H, DND-H, snr-DND-H. 

In addition to the diamond peak, all the spectra contain a D-band peak at 1350–1360 cm^−1^ and G-band peak between 1500–1700 cm^−1^ corresponding to the C sp^2^ carbon phase (defects and disordered C sp^2^ phase, grey rectangular). Sputtering induced a reduction of C sp^3^ peaks and an increase of D-band and G-band peaks in all the samples. In DND samples a redshift of the G-band was observed which indicates a change in the C sp^2^ carbon surface arrangement, probably from more or less isolated defects to more continuous domains or patches [53].

Raman spectroscopy and XPS measurements have confirmed the dependence of the graphitization process on the type and size of NDs. Previous studies have attributed the temperature-induced graphitization of NDs to a reduction in their surface energy [54], achieved by the formation of fullerene-like or bucky diamond C sp^2^ structures on the H-free C sp^3^ ND surfaces [55]. This C sp^2^-type surface reconstruction eliminates the unfavorable surface dangling bonds on clean ND surfaces. Therefore, a sequence involving ND surface cleaning or purification, H desorption, and the formation of surface dangling bonds is expected to precede the C sp^3^ to C sp^2^ phase transition, i.e., ND graphitization.

Our comparative study of NDs with different sizes, ratios of defects, contamination, and initial C sp^2^ concentrations indicates that initial defect density in NDs plays a significant role in the graphitization process. The well-ordered HPHT ND-H particles demonstrated greater resistance to annealing, surface functional group desorption, followed by graphitization. Conversely, defects and potential strain in smaller DNDs could facilitate desorption of hydrogen or related moieties at lower temperatures. Consequently, smaller, and defective NDs may exhibit higher graphitization rates, potentially leading to the formation of multiple graphitic (onion-like) C sp^2^ shells on C sp^3^ DND cores, rather than C sp^2^-type surface reconstructions.

## 4. Conclusions

We investigated the graphitization of three different types of hydrogenated NDs (top-down or bottom-up HPHT, and DNDs) using XPS, UPS, and Raman spectroscopy. To accurately assign the C sp3 and C sp2 phase features in the ND samples, we first collected referential photoelectron spectra of hydrogenated monocrystalline diamond C(111)-H and HOPG. Utilizing the temperature-dependent in situ XPS measurements, we observed that the graphitization onset temperatures for employed NDs varied. In particular, the graphitization onset temperatures decreased with decreasing ND size; >700 °C for ~18 nm HPHT ND-H, >600 °C for 5 nm DND-H and 2–3 nm snr-DND-H. Additionally to the sp^2^-related component areas in the C 1s peaks, we observed signs of graphitization such as the broadening of the sp-peak in the valence band, an increase in the π-band peak intensity, and a decrease in the 1332 cm^−1^ Raman diamond peak intensity. The C sp^2^ related contributions in the spectra and the correct assignment were independently verified through the ND surface amorphization after ArCIB sputtering. We found that the energy separation between the C 1s sp^3^ peak and valence band sp-peak remained constant at 271.3 ± 0.1 eV, enabling clear assignment of the sp^2^ and sp^3^ components in the C 1s spectra as well as calibration of UPS spectra. Overall our findings suggest that the combination of XPS, VB-XPS and UPS can help to resolve uncertainties in identifying of sp^2^ or sp^3^ components in C 1 s peak, and provide a solid basis for qualitative and quantitative investigation of ND modifications, here demonstrated on ND graphitization.

## Figures and Tables

**Figure 1 nanomaterials-14-00590-f001:**
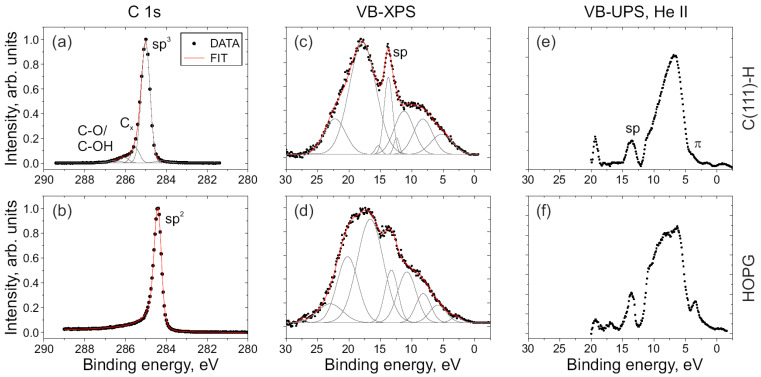
Photoelectron emission spectra from the hydrogenated monocrystalline C(111)-H diamond (upper row) and HOPG (bottom row) surfaces. (**a**,**b**) High-resolution C 1s XPS spectra, (**c**,**d**) valence band XPS spectra, (**e**,**f**) valence band UPS He II spectra. Peak components are indicated by thin grey lines. sp^3^ and sp^2^ components are shifted with respect to each other, whereas sp-band positions are similar for all samples.

**Figure 2 nanomaterials-14-00590-f002:**
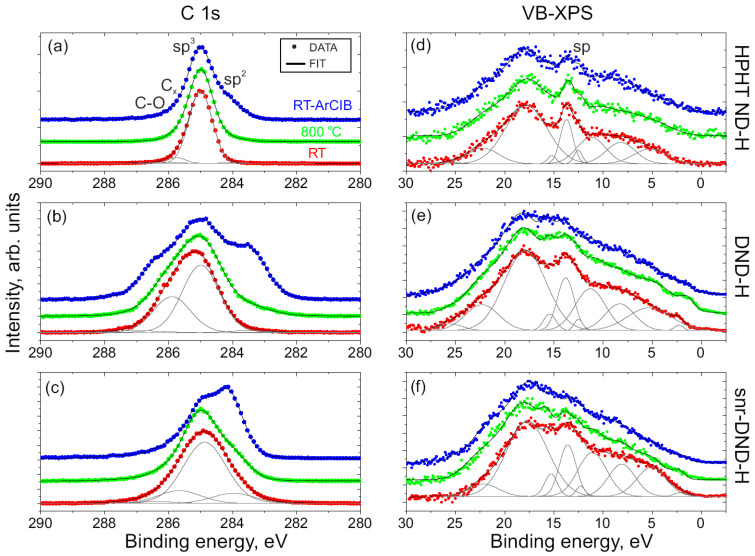
(**a**–**c**) XPS high-resolution C 1s and (**d**–**f**) valence band spectra of HPHT ND-H, DND-H, and snr-DND-H samples. Samples were measured at room temperature (as-prepared, RT), annealed at 800 °C in UHV, and sputtered by ArCIB at RT. Peak components are shown by grey lines. An increase in the sp^2^ phase of carbon has been observed after sample annealing and sputtering. The sp-peak in the valence band is smeared out after graphitization.

**Figure 3 nanomaterials-14-00590-f003:**
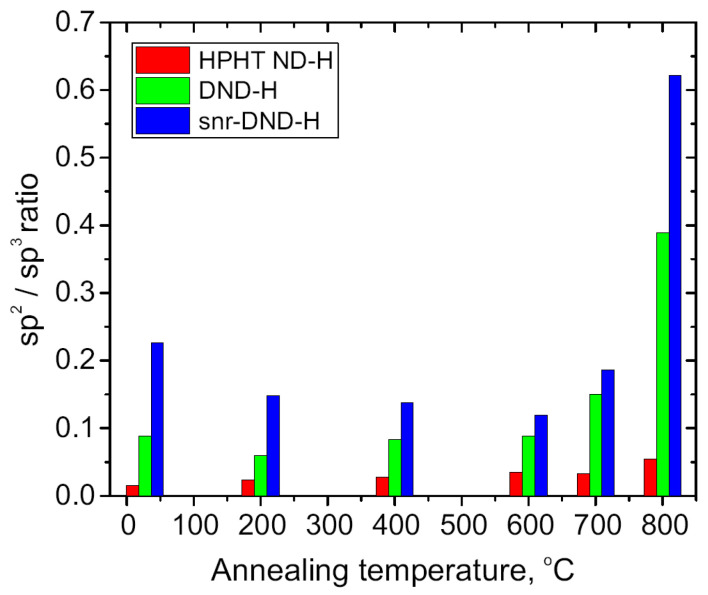
Dependence of the sp^2^/sp^3^ component ratio in the C 1s peak on the annealing temperature. An increase in sp^2^ phase concentration was observed at temperatures above 700 °C and 600 °C for HPHT ND-H and DND-H, SND-H samples, respectively.

**Figure 4 nanomaterials-14-00590-f004:**
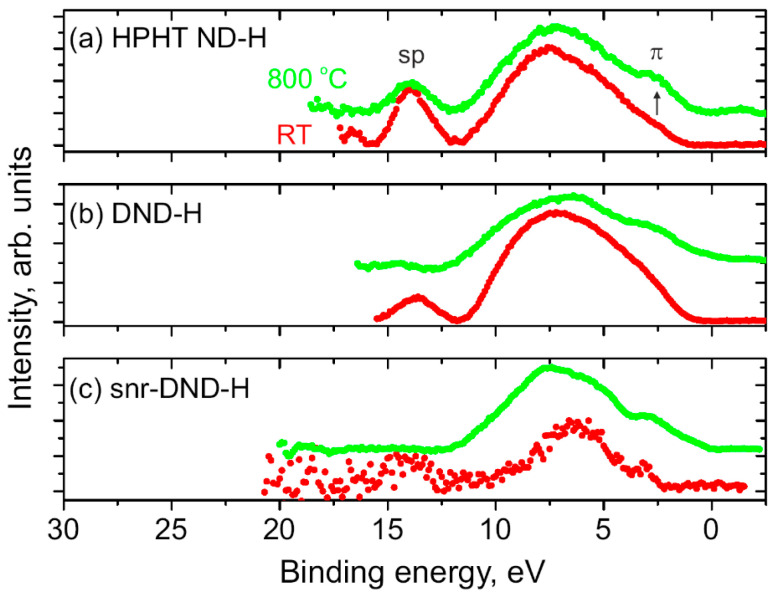
UPS He II spectra of (**a**) HPHT ND-H, (**b**) DND-H, and (**c**) snr-DND-H samples measured at RT (red curves) and after annealing at 800 °C (green curves). The intensity of the sp peak decreased, and the intensity of the π peak increased after annealing (graphitization).

**Figure 5 nanomaterials-14-00590-f005:**
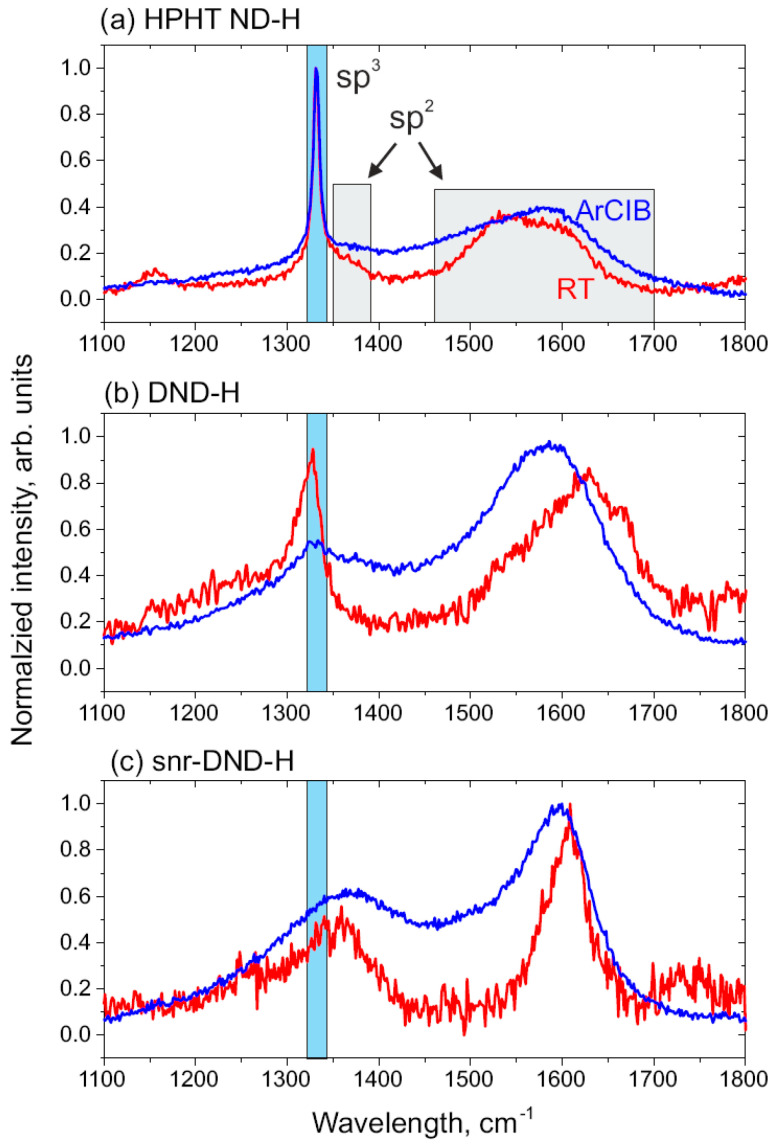
Raman spectra of as-prepared and ArCIB sputtered (**a**) HPHT ND-H, (**b**) DND-H, and (**c**) snr-DND-H samples. All samples were measured at RT. Sputtering induced disorder on the surface and broadening of the diamond peaks as well as the sp^2^-related peaks (D-bands, G-bands).

## Data Availability

Data are contained within the article and Appendix A.

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
