# Peer review of "Utilizing Constant Energy Difference between sp-Peak and C 1s Core Level in Photoelectron Spectra for Unambiguous Identification and Quantification of Diamond Phase in Nanodiamonds"

_nanomaterials, 2024, doi:10.3390/nano14070590_

Round 1
Reviewer 1 Report
Comments and Suggestions for Authors
This manuscript reports a study on thermal annealing of various nanodiamond samples in vacuum. The authors studied the process by in-situ XPS, UPS and Raman Spectroscopy. They also proposed a method of using the binding enregy difference between C 1s (sp3) and the valance band sp peak for precisely determination of binding energy shifts. In general, this paper is clearly organzied and well-written. I suggest the authors slightly revise their paper for better understanding of the results.
1. Preparation of samples for XPS and UPS measurements should be discussed in the manuscript. For measurements of nano powderss, there are different methods to fix them on sample holders. One typical way is to push it into a soft indium metal. Or it can be coated on a conductive tape. The method of sample preparation influence the characterization results.
2. Why the authors selected He II for UPS?
3. There are information depth difference between each peaks as well as XPS/UPS. This might be observed especially on sputtered surface. Did the authors observe such effects in their comprasion?
Comments on the Quality of English Language
English is acceptable but can be further improved.
Author Response
Responses are included in the attached Report Note file for the Reviewer #1.

Reviewer 2 Report
Comments and Suggestions for Authors
I do believe that the authors of the paper entitled "Utilizing Constant Energy Difference between sp-peak and C 1s Core Level in Photoelectron Spectra for Unambiguous Identification and Quantification of Diamond Phase in Nanodiamonds" have done an excellent work in this paper. The article is very interesting and irá gives valuable information to the field. Therefore, I recommend to publish the paper in tis actual form.
Author Response
Thank you very much for your report and positive evaluation of our work. No revision or answers to comments were requested.
Reviewer 3 Report
Comments and Suggestions for Authors
The work submitted by Romanyuk et al. is a really interesting and valuable study of identification and quantification of diamond phase in nanodiamonds. The authors suggests the metodology based on the analysis of sp-peak in the valence band spectra (at around 13.7 eV) and the sp3 component in the C 1s core level spectra (at 285.0 eV) obtained by XPS. In present work three types of of hydrogenated NDs were investigated by the fruitful combination of different techniques as in situ XPS, in situ UPS and Raman spectroscopy. It should be mentioned that the authors provide the validation of the results obtained for those samples by studying the reference samples as hydrogenated single crystal C(111)-H and highly ordered pyrolytic graphite (HOPG). Finally the proposed methodology which is used the combination of XPS, VB-XPS and UPS can help to resolve uncertainties in identifying of sp2 or sp3 components in C1 s peak, and provide a solid basis for qualitative and quantitative investigation of ND modifications, here demonstrated on ND graphitization. It should be mentioned that the manuscript is well-written, the introduction clearly lays out reasons for current study. The work has been carefully done and the results sound. From my opinion the manuscript can be accepted for publication in Nanomaterials as it is, with no additional actions from the Authors.
Author Response

(The authors gave the same response as above.)
